# Few-Shot Non-Parametric Learning with Deep Latent Variable Model

**Zhiying Jiang**[1,2]    **Yiqin Dai**[2]    **Ji Xin**[1]    **Ming Li**[1]    **Jimmy Lin**[1]
[1] University of Waterloo    [2] AFAIK.
zhiying.jiang@uwaterloo.ca   quinn@afaik.io
{ji.xin, mli, jimmylin}@uwaterloo.ca

## Abstract

Most real-world problems that machine learning algorithms are expected to solve face the situation with (1) unknown data distribution; (2) little domain-specific knowledge; and (3) datasets with limited annotation. We propose Non-Parametric learning by Compression with Latent Variables (NPC-LV), a learning framework for any dataset with abundant unlabeled data but very few labeled ones. By only training a generative model in an unsupervised way, the framework utilizes the data distribution to build a compressor. Using a compressor-based distance metric derived from Kolmogorov complexity, together with few labeled data, NPC-LV classifies without further training. We show that NPC-LV outperforms supervised methods on all three datasets on image classification in the low data regime and even outperforms semi-supervised learning methods on CIFAR-10. We demonstrate how and when negative evidence lowerbound (nELBO) can be used as an approximate compressed length for classification. By revealing the correlation between compression rate and classification accuracy, we illustrate that under NPC-LV how the improvement of generative models can enhance downstream classification accuracy.

## 1 Introduction

The progress of deep neural networks drives great success of supervised learning with huge labeled datasets [1, 2, 3]. However, large labeled datasets are luxurious in many applications and huge amounts of training parameters make the model easy to overfit and hard to generalize to other datasets. The urge to learn with small labeled dataset prompts Few-Shot Learning (FSL). However, most few-shot classification settings require either an auxiliary "support set" [4, 5, 6, 7] that contains $c$ classes, each with $k$ samples ($c$-way $k$-shot); or prior knowledge about the dataset, where data augmentation can be performed within the same dataset [8, 9, 10] or from other weakly-labeled/unlabeled/similar datasets [11, 12, 13]. This setting is not widely applicable to every dataset in practice, as it requires either the elaborate construction of an additional "support set" or augmentation algorithms tailored to specific datasets. Pre-trained models, on the other hand, do not require adhoc "support" and have proved to be good at few-shot learning [14] and even zero-shot learning [15]. However, thousands of millions of training parameters make the model hard to be retrained but only fine-tuned. When the data distribution is substantially different from any datasets used in pre-training, the inductive bias from pre-training holds up fine-tuning, making the model less pliable [16].

Goals from the above learning paradigms can be summarized as to design algorithms that can be applied to any dataset, and can learn with few labeled data, ideally with no training. "No Free Lunch" [17] implies that it's impossible to have an algorithm that is both "universal" and "best". But how good can a "universal" algorithm be, especially in the low data regime, with no external data resources? Specifically, we are interested in a new setting, *Non-Supported Few-Shot Learning* (NS-FSL), defined as follows:

36th Conference on Neural Information Processing Systems (NeurIPS 2022).

Given any target dataset $\mathbf{D} = (\mathbf{x}_1, \mathbf{x}_2, ..., \mathbf{x}_n)$ belonging to $c$ classes. For each class, we have $k$ labeled samples ($1 \leq k \leq 10$). The remaining $n - ck$ unlabeled samples need to be classified into $c$ classes without the need of support sets, any other datasets, or training parameters.

This setting is similar to semi-supervised learning's but excludes labeled information in training. Ravi and Larochelle [18] demonstrate that it's hard to optimize a neural network when labeled data is scarce. In order to make minimum assumptions about labeled data, we aim at using parameter-free methods. The goal is to grasp the data-specific distribution $p(\mathbf{x})$, with minimal premises on the conditional probability distribution $p(y|\mathbf{x})$. Deep generative models with explicit density estimation are perfect candidates for this goal. The problem then becomes: given trained generative models, how to take full advantage of the information obtained from them for classification? Using a latent representation only utilizes $p(\mathbf{z}|\mathbf{x})$, which just includes partial information. Even for those latent generative models that do not suffer from posterior collapse [19], $p(\mathbf{z}|\mathbf{x})$'s insufficiency for classification with non-parametric methods like $k$-nearest-neighbor is shown in both previous works [20, 21] and our experiments.

Inspired by previous work that uses compressor-based distance metrics for non-parametric learning [22, 23, 24, 25], we propose Non-Parametric learning by Compression with Latent Variables (NPC-LV), a learning framework that consists of deep-generative-model-based compressors and compressor-based distance metrics. It leverages the power of deep generative models without exploiting any label information in the probabilistic modeling procedure. With no further training, this framework can be directly used for classification. By separating probabilistic modeling from downstream tasks that require labels, we grasp the unique underlying structures of the data in every dataset, and further utilize these structures in downstream tasks with no parameter tuning needed. We view this learning framework as a *baseline* in this setting, for any dataset. We argue it is "parameter-free" as there are no parameters involved in the classification stage for labeled data. Basically it means training a generative model as is and getting a classifier for free.

Our contributions are as follows: (1) We frame existing methods into a general learning framework NPC, based on which we derive NPC-LV, a flexible learning framework with replaceable modules. (2) We use NPC-LV as a baseline for a common learning scenario NS-FSL with neither support sets nor further training. (3) Our method outperforms supervised methods by up to 11.8% on MNIST, 18.0% on FashionMNIST, 59% on CIFAR-10 on image classification in the low data regime. It outperforms non-parametric methods using latent representation on all three datasets. It even outperforms semi-supervised learning methods on CIFAR-10. (4) We show how negative evidence lowerbound (nELBO) can be used for classification under this framework. (5) We find the correlation between bitrate and classification accuracy. This finding suggests how improvements in the domain of deep-learning-based-compressors can further boost classification accuracy under this framework.

## 2 Background

### 2.1 Information Theory — Data Compression

In a compression scenario, suppose we have a sender *Alice* and a receiver *Bob*. *Alice* wants to send a message that contains a sequence of symbols $\mathbf{x} = (x_1, x_2, ..., x_n)$ to *Bob*. The ultimate goal of the lossless compressor is to compress $\mathbf{x}$ into the minimum amount of bits $\mathbf{x}'$ that can later be decompressed back to $\mathbf{x}$ by *Bob*. To achieve the shortest compressed length, shorter codes are assigned to symbols with higher probability. According to Shannon's Source Coding Theorem [26], this length of bits is no shorter than the entropy of the sequence, whose definition is $H(\mathbf{x}) \triangleq \mathbb{E}[-\log p_{\text{data}}(\mathbf{x})]$, where $p_{\text{data}}(\mathbf{x})$ represents the probability distribution of each symbol in the sequence. Instead of coding symbols one by one, stream code algorithms like Asymmetric Numeral Systems (ANS) [27] convert $\mathbf{x}$ to a sequence of bits $\mathbf{x}'$ and reaches this optimal code length for the whole sequence with an overhead of around 2 bits, given $p_{\text{data}}(\mathbf{x})$. However, the "true" probabilistic distribution $p_{\text{data}}(\mathbf{x})$ is unknown to us. We can only access samples and approximate it with $p_\theta(\mathbf{x})$. That is:

$$\mathbb{E}[-\log p_\theta(\mathbf{x})] \geq H(\mathbf{x}) \triangleq \mathbb{E}[-\log p_{\text{data}}(\mathbf{x})]. \tag{1}$$

Given an entropy coding scheme, the better $p_\theta(\mathbf{x})$ approximates $p_{\text{data}}(\mathbf{x})$, the closer we can get to the minimum code length. Modeling $p_\theta(\mathbf{x})$ is where deep generative model with density estimation can help. Possible coding schemes and generative models for compressors are discussed in 3.2.

## 2.2 Algorithmic Information Theory — Kolmogorov Complexity and Information Distance

While information theory is built on data distributions, algorithmic information theory considers "single" objects without the notion of a probability distribution. Kolmogorov complexity $K(x)$ [28] is used to describe the length of the shortest binary program that can produce $x$ on a universal computer, which is the ultimate lower bound of information measurement. Similarly, the Kolmogorov complexity of $x$ given $y$ is the length of the binary program that on input $y$ outputs $x$, denoted as $K(x|y)$. Based on Kolmogorov complexity, Bennett et al. [29] derive *information distance $E(x, y)$*:

$$E(x, y) = \max\{K(x|y), K(y|x)\} = K(xy) - \min\{K(x), K(y)\}.^1 \tag{2}$$

The idea behind this measurement, at a high level, is that the similarity between two objects indicates the existence of a simple program that can convert one to another. The simpler the converting program is, the more similar the objects are. For example, the negative of an image is very similar to the original one as the transformation can be simply described as "inverting the color of the image".

**Theorem 1.** *The function $E(x, y)$ is an admissible distance and a metric. It is minimal in the sense that for every admissible distance $D$, we have $E(x, y) \leq D(x, y) + O(1)$.*

Intuitively, *admissible distance* refers to distances that are meaningful (e.g., excluding metrics like $D(x, y) = 0.3$ for any $x \neq y$) and computable (formal definition is in Appendix B). Combining these definitions, we can see Theorem 1 means $E(x, y)$ is *universal* in a way that it is optimal and can discover all effective similarities between two objects (proof is shown in Appendix H).

In order to compare the similarity, relative distance is preferred. Li et al. [30] propose a normalized version of $E(x, y)$ called *Normalized Information Distance* (NID).

**Definition 1** (**NID**). *NID is a function: $\Omega \times \Omega \to [0, 1]$, where $\Omega$ is a non-empty set, defined as:*

$$NID(x, y) = \frac{\max\{K(x|y), K(y|x)\}}{\max\{K(x), K(y)\}}. \tag{3}$$

Equation (3) can be interpreted as follows: Given two sequences $x, y$, $K(y) \geq K(x)$:

$$\text{NID}(x, y) = \frac{K(y) - I(x : y)}{K(y)} = 1 - \frac{I(x : y)}{K(y)}, \tag{4}$$

where $I(x : y) = K(y) - K(y|x)$ means the *mutual algorithmic information*. $\frac{I(x:y)}{K(y)}$ means the shared information (in bits) per bit of information contained in the most informative sequence, and Equation (4) here is a specific case of Equation (3). Theoretically, NID is a desirable distance metric as it satisfies the metric (in)equalities (definition in Appendix B) up to additive precision $O(1/K(\cdot))$ where $K(\cdot)$ is the maximum complexities of objects involved in (in)equalities (proof shown in [31]).

## 3 Non-Parametric learning by Compression with Latent Variables

Non-Parametric learning by Compression (NPC) consists of three modules — a distance metric, a compressor and an aggregation method shown in Figure 1. NPC-LV leverages NPC by including neural compressors based on deep generative models. We introduce the derivation of compressor-based distance metrics in Section 3.1; the generative-model-based compressor in Section 3.2; an integration of this framework with generative models in Section 3.3.

### 3.1 Compressor-based Distance Metric

Universal as NID is, it is uncomputable, because Kolmogorov complexity is uncomputable. Cilibrasi and Vitányi [32] propose *Normalized Compression Distance* (NCD), a quasi-universal distance metric based on real-world compressors. In this context, $K(x)$ can be viewed as the length of $x$ after being maximally compressed. Suppose we have $C(x)$ as the length of compressed $x$ produced by a real-world compressor, then NCD is defined as:

$$\text{NCD}(x, y) = \frac{C(xy) - \min\{C(x), C(y)\}}{\max\{C(x), C(y)\}}. \tag{5}$$

---

[1]Note that $K(xy)$ denotes the length of the shortest program computing $x$ and $y$ without telling which one is which (i.e., no seperator encoded between $x$ and $y$).

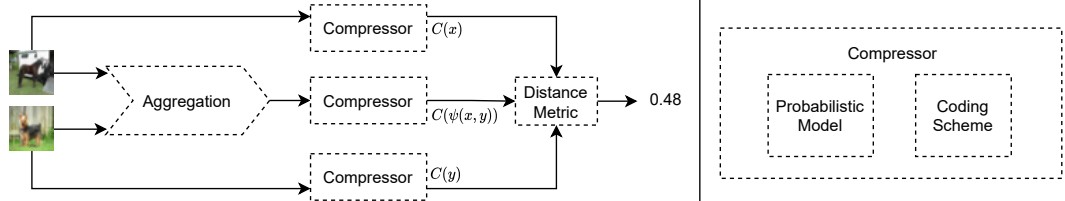

Figure 1: NPC framework with trainable deep probabilistic models. Replaceable modules are indicated with dashed lines.

The better the compressor is, the closer NCD approximates NID. With a *normal* compressor (discussed in detail in Appendix B), NCD has values in [0,1] and satisfies the distance metric (in)equalities up to $O(\log n/n)$ where $n$ means the maximum binary length of a string involved [33]. NCD is thus computable in that it not only uses compressed length to approximate $K(x)$ but also replaces conditional Kolmogorov complexity with $C(xy)$ that only needs a simple concatenation of $x, y$. Li et al. [31] simplify NCD by proposing another *compression-based dissimilarity measurement* (CDM):

$$\text{CDM}(x, y) = \frac{C(xy)}{C(x) + C(y)}. \tag{6}$$

Chen et al. [24] use another variation ranging from 0 to 1:

$$\text{CLM}(x, y) = 1 - \frac{C(x) + C(y) - C(xy)}{C(xy)}. \tag{7}$$

NCD, CDM and CLM are different variations of Kolmogorov-based distance metrics. We empirically evaluate their performance in Section 5.

### 3.2 Trained Generative Models as Compressors

Previous works [24, 29, 32] demonstrate the success of compression-based distance metrics in sequential datasets like time series, DNA, and texts using non-neural compressors like gzip, bzip2, PPMZ. Deep-generative-model–based compressors can take Non-Parametric learning by Compression (NPC) to the next level by expanding to more data types using better compressors. We mainly focus on variational autoencoder (VAE) based compressors with a brief introduction to other neural compressors.

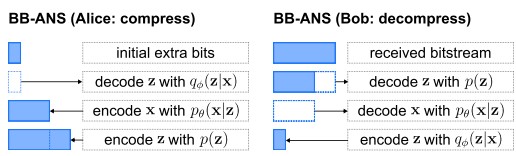

Figure 2: BB-ANS compress & decompress

**VAE Family:** The relation between VAE and "bits-back" has been revealed in multiple previous works [34, 35]. Townsend et al. [36] use latent variable models by connecting Asymmetric Numeral Systems (ANS) [27] to the "bits-back" argument [37] (BB-ANS). In the setting of the "bits-back argument", we assume *Alice* has some extra bits of information to send to *Bob* alongside **x**. It's also assumed that both *Alice* and *Bob* have access to $p(\mathbf{z})$, $p_\theta(\mathbf{x}|\mathbf{z})$ and $q_\phi(\mathbf{z}|\mathbf{x})$ where **z** is the latent variable; $p(\mathbf{z})$ is the prior distribution of **z**; $p_\theta(\mathbf{x}|\mathbf{z})$ represents a generative network and $q_\phi(\mathbf{z}|\mathbf{x})$ represents an inference network. As shown in Figure 2, *Alice* first decodes the extra information according to $q_\phi(\mathbf{z}|\mathbf{x})$ to generate a sample **z**.[2] **z** is further used to encode **x** with $p_\theta(\mathbf{x}|\mathbf{z})$ and **z** itself is encoded using $p(\mathbf{z})$. *Bob* then reverses this procedure and recovers the extra bits by encoding with $q_\phi(\mathbf{z}|\mathbf{x})$. For a single data point, the length of the final bitstream is:

$$N = n_{\text{extra}} + \log q_\phi(\mathbf{z}|\mathbf{x}) - \log p_\theta(\mathbf{x}|\mathbf{z}) - \log p(\mathbf{z}). \tag{8}$$

We can see the expectation of $N - n_{\text{extra}}$ is equal to the negative evidence lower bound (nELBO):

---

[2]Note that "encoding", "decoding" here follows data compression's convention instead of variational autoencoder's.

$$\mathbb{E}_{q_\phi(\mathbf{z}|\mathbf{x})}[N - n_{\text{extra}}] = -\mathbb{E}_{q_\phi(\mathbf{z}|\mathbf{x})} \log \frac{p_\theta(\mathbf{x}, \mathbf{z})}{q_\phi(\mathbf{z}|\mathbf{x})} = -\text{ELBO} \tag{9}$$

ELBO above is derived from the "bits-back argument" in the context of compression. Now, from the perspective of latent variable models like VAE, the derivation often starts from the fact that $p_\theta(\mathbf{x}) = \int p_\theta(\mathbf{x}|\mathbf{z})p(\mathbf{z})$ is intractable. $q_\phi(\mathbf{z}|\mathbf{x})$ is then introduced as an inference model to approximate $p(\mathbf{z}|\mathbf{x})$ in order to work around the intractability problem, which brings up the marginal log likelihood:

$$\log p_\theta(\mathbf{x}) = \mathbb{E}_{q_\phi(\mathbf{z}|\mathbf{x})} \log \frac{p_\theta(\mathbf{x}, \mathbf{z})}{q_\phi(\mathbf{z}|\mathbf{x})} + \mathbb{E}_{q_\phi(\mathbf{z}|\mathbf{x})} \log \frac{q_\phi(\mathbf{z}|\mathbf{x})}{p(\mathbf{z}|\mathbf{x})},$$
$$\text{ELBO} = \log p_\theta(\mathbf{x}) - D[q_\phi(\mathbf{z}|\mathbf{x})\|p(\mathbf{z}|\mathbf{x})]. \tag{10}$$

We only need to optimize the lower bound, as minimizing nELBO means maximizing $\log p_\theta(\mathbf{x})$ — the likelihood of generating real data and minimizing KL divergence between $q_\phi(\mathbf{z}|\mathbf{x})$ and $p(\mathbf{z}|\mathbf{x})$ at the same time, which is the same objective function from what we derive using "bits-back".

This equivalence demonstrates that an optimized latent variable model can be used directly for compression as, from the data compression perspective, it minimizes the code length attainable by bits-back coding using the model. With the help of ANS, we can encode symbols into bitstreams or decode bitstreams back to symbols with trained latent variable models. Details of ANS and discretizing continuous latent variables are shown in Appendix D and E.

**ARM:** Autoregressive models (ARMs) model $p(\mathbf{x})$ as: $p(\mathbf{x}) = p(x_0) \prod_{i=1}^n p(x_i|\mathbf{x}_{i-1})$. The exact likelihood estimation makes it capable of lossless compression. But instead of using ANS, which is a stack-like coding scheme, queue-like ones (e.g., Arithmetic Coding (AC) [38]) should be used. Computational inefficiency is the main drawback of ARMs such as RNN [39], but causal convolutions [40, 41] can alleviate the problem.

**IDF:** Integer Discrete Flow (IDF) [42] can also optimize towards the exact log-likelihood. Similar to other flow-based models, it utilizes an invertible transformation $f$, but works on discrete distributions with additive coupling and rounding operations. For IDF, ANS can be used as the entropy coder.

Our experiments and discussion are mainly about VAE-based compressors as their architectures can be flexibly changed under the "bits-back" argument.

### 3.3 NPC-LV

We've shown in Section 3.2 that we can plug in any trained latent variable model in exchange for a near optimal compressor under the framework of BB-ANS. To show that the coding scheme is replaceable, we introduce a coding scheme variation, Bit-Swap [43], in the following experiments. The difference between BB-ANS and Bit-Swap is the encoding and decoding order when there is more than one latent variable. A detailed comparison is shown in Appendix F. The generative model we use for both BB-ANS and Bit-Swap is a hierarchical latent generative model (details are in Appendix G), also known as Deep Latent Gaussian Model (DLGM) [44].

**Aggregation:** In addition to coding schemes and probabilistic models for compressors, aggregation methods can also be replaced. Previous works [32] assume $xy$ in $C(xy)$ means the "concatenation" of two inputs. We expand this assumption to a more general case where "aggregation" can be another kind of aggregation function, represented as $C(\psi(x, y))$ in Figure 1. We justify this generalization as changing aggregations methods may make compressor-based distance metrics *not admissible* (shown in Appendix B). More sophisticated strategies of aggregation are left to future work. We also discuss other replaceable modules in detail in Appendix A.

We use BB-ANS with NCD as a concrete instance to demonstrate this framework on a classification task shown in Algorithm 1. $\mathbf{D}_{\text{train}}^U$ and $\mathbf{D}_{\text{train}}^L$ mean the unlabeled and labeled training sets; predefined functions are in teal. The algorithm can be simplified into four steps: (1) Train a VAE on the unlabeled training dataset; (2) Apply ANS with discretization on the trained VAE to get a compressor; (3) Calculate the distance matrix between pairs $(\mathbf{x}_{\text{test}}, \mathbf{x}_{\text{train}})$ with the compressor and NCD; (4) Run $k$-Nearest-Neighbor($k$NN) classifier with the distance matrix.

**nELBO as estimated compressed length:** Specifically for VAE-based compressors, nELBO is the estimated length of the compressed bitstream as Equation (9) shows. Therefore, we can use it directly

---

**Algorithm 1** NPC-LV (use VAE and NCD as an example)

---

**Input:** $k$, $\mathbf{D}_{\text{test}}$, $\mathbf{D}_{\text{train}} = \{\mathbf{D}_{\text{train}}^U, \mathbf{D}_{\text{train}}^L\}$, $\mathbf{D}_{\text{train}}^L = \{X^L, y^L\}$
trained_vae = trainVAE($\mathbf{D}_{\text{train}}^U$)
vae_compressor = ANS(trained_vae)
**for** $\mathbf{x}_{\text{test}}$ in $\mathbf{D}_{\text{test}}$ **do**
    C_$\mathbf{x}_{\text{test}}$ = len(vae_compressor($\mathbf{x}_{\text{test}}$))
    distances = []
    **for** $\mathbf{x}_{\text{train}}$ in $X^L$ **do**
        *// Calculate NCD distance with $C(x)$, $C(y)$, and $C(\psi(x, y))$*
        C_$\mathbf{x}_{\text{train}}$ = len(vae_compressor($\mathbf{x}_{\text{train}}$))
        C_agg = len(vae_compressor( aggregate($\mathbf{x}_{\text{test}}$, $\mathbf{x}_{\text{train}}$))
        NCD = (C_agg $-$ min\{C_$\mathbf{x}_{\text{test}}$, C_$\mathbf{x}_{\text{train}}$\}) / max\{C_$\mathbf{x}_{\text{test}}$, C_$\mathbf{x}_{\text{train}}$\}
        distances = push(NCD, distances)
    **end for**
    k_nearest_indicies = argsort(distances)[:$k$]
    $y_{\text{test}}$ = majority($\{y_i^L, i \in$ k_nearest_indicies$\}$)
**end for**

---

without actual compression. This can further simplify our method as we don't need to discretize continuous distributions or apply an entropy coder.

The reason why the underlying data distribution can help the classification is based on the *manifold assumption*, which is a common assumption in SSL [45]. It states that the input space consists of multiple low-dimensional manifolds, and data points lying on the same manifold have the same labels. This assumption helps alleviate the curse of dimensionality and may explain the effectiveness of using $k$NN with few labeled data. Due to the fact that our training process does not use labeled data, our method does not rely on other common assumptions in SSL like the *smooth assumption* and the *low-density assumption*. The facts that NPC-LV makes very few assumptions about datasets and that compressors are data-type-agnostic make this framework extensible to other data types beyond images. For example, the combination of an autoregressive model (e.g., character recursive neural network) and arithmetic coding [46] can be used in our framework for sequential data.

## 4 Related Work

**Non-parametric learning with *Information Distance*:** Bennett et al. [29] propose *information distance* as a universal metric, based on which several papers [47, 48] propose more fine-grained distance metrics. Chen et al. [24], Li et al. [31], Cilibrasi and Vitányi [32] derive more practical distance metrics based on real-world compressors. Empirical results [22, 23, 24, 25] show that even without any training parameters, those compressor-based distance metrics can produce an effective distance matrix for clustering and classification on time series datasets. Cilibrasi and Vitányi [32] further push this direction to more types of datasets, including images that are represented in "#" (black pixel) and "." (white pixel). We unify previous work in the NPC framework, expand it to real image datasets and leverage it with neural compressors.

**Compression:** Shannon [26] establishes source coding theorems, showing that entropy rate is the limit of code rate for lossless compression. Huffman Coding [49] achieves the optimal symbol code whose length is upper-bounded by $H(\mathbf{x}) + 1$ per symbol. The 1 bit overhead is due to the fact that $-\log p(x)$ is not always an integer. Stream coding techniques like AC [38] and ANS [27] further optimize by representing the whole message with numeral systems. These entropy coders then can be combined with probabilistic modeling using neural networks [46, 50, 51] and used in our framework.

**Semi-Supervised Learning with VAE:** The evaluation paradigm in this paper is closest to Semi-Supervised Learning (SSL). Kingma et al. [52] design two frameworks for utilizing autoencoders in downstream classification tasks. The first (M1) is to train a tSVM [53] with latent representation output by a trained VAE. The second (M2) is to train a VAE with labels as another latent variable. M1 only requires a standard VAE but tSVM suffers from optimization difficulty [54], making it hard to be generally applicable for VAE. More recent VAE-based methods [55, 56] are built on M2. These methods don't train a generative model in an unsupervised way like we do.

| Data | MNIST | | | FashionMNIST | | | CIFAR-10 | | |
|---|---|---|---|---|---|---|---|---|---|
| #Shot | 5 | 10 | 50 | 5 | 10 | 50 | 5 | 10 | 50 |
| **Supervised Learning** | | | | | | | | | |
| SVM | 69.4±2.2 | 77.1±1.5 | 87.6±0.4 | 67.1±2.1 | 71.0±1.6 | 78.4±0.5 | 21.1±1.9 | 23.6±0.5 | 27.2±1.2 |
| #Param | 35,280 | | | 35,280 | | | 105,840 | | |
| CNN | 72.4±3.5 | 83.7±2.6 | 93.2±2.8 | 67.4±1.9 | 70.6±2.5 | 80.5±0.7 | 23.4±2.9 | 28.3±1.9 | 38.7±1.9 |
| #Param | 1,199,882 | | | 1,199,882 | | | 1,626,442 | | |
| VGG | 69.4±5.7 | 83.9±3.2 | 94.4±0.6 | 62.8±4.1 | 70.5±4.5 | 81.5±1.1 | 22.2±1.6 | 29.7±1.8 | 42.6±1.2 |
| #Param | 28,148,362 | | | 28,148,362 | | | 28,149,514 | | |
| **Semi-Supervised Learning** | | | | | | | | | |
| VAT | 97.0±0.3 | 97.4±0.1 | 98.4±0.1 | *74.1±0.8* | 78.4±0.3 | 87.1±0.2 | 25.4±2.0 | 27.8±4.2 | 60.9±6.1 |
| #Param | 1,469,354 | | | 1,469,354 | | | 1,469,642 | | |
| MT | 78.4±2.0 | 82.8±1.9 | 98.6±0.2 | 58.1±2.8 | 70.8±0.8 | 87.1±0.1 | 31.7±1.5 | 35.9±1.1 | *64.3±1.6* |
| #Param | 1,469,354 | | | 1,469,354 | | | 1,469,642 | | |
| **Non-Parametric Learning** | | | | | | | | | |
| Single | 65.6±1.2 | 76.8±0.8 | 86.3±0.3 | 40.2±1.4 | 53.4±1.1 | 70.0±0.4 | 17.3±0.9 | 19.2±0.7 | 23.4±0.3 |
| #Param | 0 | | | 0 | | | 0 | | |
| Hier | 73.6±3.1 | 82.3±2.1 | 90.4±1.4 | 69.5±3.5 | 72.5±1.9 | 78.7±1.3 | 22.2±1.6 | 24.2±4.9 | 26.2±2.9 |
| #Param | 0 | | | 0 | | | 0 | | |
| **Non-Parametric learning by Compression with Latent Variables (NPC-LV)** | | | | | | | | | |
| nELBO | **75.2±1.5** | 81.4±1.1 | 91.0±1.0 | **72.2±2.2** | 76.7±1.5 | 85.6±1.1 | __34.1±1.8__ | **34.6±2.0** | 35.6±2.5 |
| #Param | 0 | | | 0 | | | 0 | | |
| Bit-Swap | **75.7±3.6** | 83.3±0.9 | 90.9±0.2 | **73.5±3.7** | **76.0±1.4** | 82.6±1.2 | **32.2±3.5** | **32.8±1.9** | 35.7±1.1 |
| #Param | 0 | | | 0 | | | 0 | | |
| BB-ANS | **77.6±0.4** | **84.6±2.1** | 91.4±0.6 | *74.1±3.2* | **77.2±2.2** | **83.2±0.7** | *__35.3±2.9__* | *__36.0±1.8__* | 37.4±1.2 |
| #Param | 0 | | | 0 | | | 0 | | |

Table 1: Test accuracy of methods with number of learnable parameters for classification. #Shot refers to the the number of training samples per class. Results report means and the 95% confidence interval over five trials. Note that "#Param" refers to parameters specifically for supervised training.

**Few-Shot Learning:** Similar to our setting, FSL also targets the low labeled data regime. Many previous papers [4, 5, 6, 7] on FSL are based on meta-learning, where the model is fed with an extra labeled support set, in addition to the target dataset. Another line of work [8, 9, 10, 11, 12, 13] utilize data augmentation. Although some of them do not require extra datasets [8, 9, 10], the augmentation algorithms can hardly be applied to every other dataset [57]. Metric-based methods [58] utilize distance metrics for FSL. But instead of modeling the probability distribution of a dataset, they model the "distance" between any pair of data points with a neural network, which still requires many labeled data during "pre-training". Our work is similar to metric-based methods in that both have the essence of nearest-neighbor. The difference is that in the "pre-training" stage, our model is not trained to learn the distance but to reconstruct the image as all standard generative models do. More importantly, we use no labeled data in this stage.

## 5 Experiments

We compare our method with supervised learning, semi-supervised learning, non-parametric learning and traditional Non-Parametric learning by Compression (NPC) on MNIST, FashionMNIST and CIFAR-10 [59, 60, 61]. For each dataset, we first train a hierarchical latent generative model with *unlabeled* training sets. During the stage of calculating the distance metric using compression, we pick 1,000 samples from the test set, due to the cost of compression and pair-wise computation, together with $n = \{5, 10, 50\}$ labeled images per class from the training set. We also report the result for $n = 50$ although it is beyond our setting. We keep the selected dataset the same for every method compared. We use **bold** to highlight the cases we outperform supervised methods, use underline to highlight the case we outperform SSL, and use *italic* to highlight the highest accuracy among all methods for reference. We perform compression with two coding schemes (BB-ANS and Bit-Swap) and also use nELBO for compressed length directly.

**Comparison with Supervised Learing:** Supervised models are trained on $10n$ labeled data. In Table 1, when $n = 50$, CNN and VGG surpass NPC-LV. In the cases where the number of labeled data is extremely limited (e.g., 5, 10 labeled data points per class), however, the BB-ANS variant

|  | MNIST | | | FashionMNIST | | | CIFAR-10 | | |
|---|---|---|---|---|---|---|---|---|---|
|  | NCD | CLM | CDM | NCD | CLM | CDM | NCD | CLM | CDM |
| gzip | 86.1 | 85.6 | 85.6 | 81.7 | 82.6 | 82.6 | 31.3 | 30.3 | 30.3 |
| bz2 | 86.8 | 86.4 | 86.4 | 81.7 | 79.0 | 79.0 | 28.0 | 27.5 | 27.5 |
| lzma | 87.4 | 88.5 | 88.5 | 80.6 | 82.7 | 82.7 | 31.4 | 30.0 | 30.0 |
| WebP | 86.4 | 87.9 | 87.9 | 69.9 | 67.3 | 67.3 | 33.3 | 34.2 | 34.2 |
| PNG | 86.8 | 89.1 | 89.1 | 74.8 | 76.9 | 76.9 | 32.2 | 28.9 | 28.9 |
| BitSwap | 93.2 | 90.9 | 93.2 | 84.3 | *84.0* | *84.0* | 36.9 | 36.9 | 36.9 |
| BBANS | *93.6* | *93.4* | *93.4* | *84.5* | 83.6 | 83.6 | *40.2* | *40.8* | *40.8* |

Table 2: Classification accuracy using different compressors and distance metrics.

outperforms all methods in every dataset. For MNIST, both BB-ANS and Bit-Swap produce more accurate results than supervised methods on 5-shot experiments; BB-ANS performs slightly better than the supervised methods in the 10-shot scenario. For FashionMNIST, all three variants outperform in 5-shot, 10-shot, and even 50-shot settings. For CIFAR-10, given 10 labeled data points per class, NPC-LV surpasses the accuracy of CNN by 27.2% and improves the accuracy of VGG by 21.2%. This enhancement is more significant in the 5-shot setting: NPC-LV improves the accuracy of CNN by 50.9% and by 59.0% for VGG. In general, we can see as the labeled data become fewer, NPC-LV becomes more advantageous.

**Comparison with Semi-Supervised Learning:** The input of NS-FSL is similar to SSL in that both unlabeled data and labeled data are involved. The difference lies in the fact that (1) our training *doesn't* use any labeled data and is purely unsupervised; (2) we only need to train the model *once*, while SSL needs to retrain for different $n$; (3) we use fewer labeled data points, which is a more practical setting for real-world problems. We choose strong semi-supervised methods that make few assumptions about the dataset. We use consistency regularization methods instead of pseudo-labeling ones as pseudo-labeling methods often assume that decision boundary should pass through low-density region of the input space (e.g., Lee [62]). Specifically, we choose MeanTeacher ("MT") [63] and VAT [64]. The core of both is based on the intuition that realistic perturbation of data points shouldn't affect the output. We train both models with $n$ labeled samples per class together with an unlabeled training set (training details are shown in Appendix C).

As we can see, NPC-LV achieves higher accuracy for CIFAR-10 in the low data regime, has competitive result on FashionMNIST, and is much lower on MNIST. The strength of our method is more obvious with more complex datasets. It's a surprising result because we do not implement any data augmentation implicitly or explicitly unlike consistency regularization methods, which utilize data perturbation and can be viewed as data augmentation. It's worth noting that on all three datasets our method using BB-ANS always outperforms ***at least one*** semi-supervised methods on the 10-shot setting, indicating that SSL methods trade "universality" for "performance" while our method is more like a baseline. Speed-wise, NPC-LV only requires training once for the generative model, and can run $k$NN on different shots ($n$) with no additional cost. In contrast, SSL methods require the whole pipeline to be retrained for every $n$.

**Comparison with Non-Parametric Learning:** In this experiment, we explore the effectiveness of using latent representations directly with $k$NN. We train the same generative model we use in NPC-LV ("Hier"), as well as a vanilla VAE with a single latent variable ("Single"). Table 1 shows that the latent representation of the vanilla one is not as expressive as the hierarchical one. Although the latent representation using the hierarchical architecture performs reasonably well and surpasses supervised methods in the 5-shot setting on MNIST and FashionMNIST, it's still significantly lower than NPC-LV in all settings. The result suggests that NPC-LV can utilize trained latent variable models more effectively than simply utilizing latent representations for classification.

**Comparison with NPC:** We investigate how Non-Parametric learning by Compression (NPC) with non-neural compressors perform with different distance metrics. We evaluate with NCD, CLM, and CDM as distance metrics, and gzip, bz2, lzma, WebP, and PNG as compressors, using 1,000 images from the test set and 100 samples per class from the training set. The result is shown in Table 2. For distance metrics, we can see CLM and CDM perform similarly well but it's not clear under what circumstances a distance metric is superior to the rest. For compressors, both Bit-Swap and BB-ANS perform much better than other compressors, indicating that generative-model-based compressors can significantly improve NPC. BB-ANS turns out to be the best compressor for classification on all three datasets.

# 6 Analyses and Discussion

## 6.1 nELBO as Compressed Length

As we've shown in Section 3.2, nELBO can be viewed as the expected length of the compressed bitstream $N - n_{\text{extra}}$. Thus, theoretically it can be used directly to approximate compressed length. In this way, we don't need to apply ANS to VAE for the actual compression, which largely simplifies the method and boosts speed. However, as we can see in Table 1, using nELBO doesn't always perform better than an actual compressor like BB-ANS. This may be because nELBO in a well-trained model regards the aggregation of two images as out-of-distribution data points; while the discretization in the actual compressor forces close probability with a certain level of precision to be discretized into the same bin, lowering the sensitivity. Better aggregation strategies need to be designed to mitigate the gap.

## 6.2 Performance Gain and Task Difficulty

We find that our method is more advantageous on more complex datasets with a lower shot number in terms of the relative performance. Specifically, we average all methods' accuracy that NPC-LV compared across different shot settings and datasets, denoted as $\bar{a}$. We then calculate the excess rate with BB-ANS variant's accuracy $b$ by $\frac{b-\bar{a}}{\bar{a}} \times 100\%$. The result is shown in Table 3.

|  | 5-shot | 10-shot | 50-shot |
|---|---|---|---|
| MNIST | 4.8% | 1.9% | -0.1% |
| FashionMNIST | 22.4% | 12.3% | 3.9% |
| CIFAR-10 | 55.8% | 38.1% | 6.3% |

Table 3: NPC-LV's excess rate compared with the average of all other methods.

The lower left part of the table represents higher task difficulty. As the shot number decreases (from right to left) and/or as the difficulty of the dataset increases (from top to bottom), our framework's performance enhancement gets higher.

## 6.3 Bitrate versus Classification Accuracy

The origin of the NPC framework comes from the intuition that the length of $x$ after being maximally compressed by a real-world compressor is close to $K(x)$. Theoretically, the closer this length approximates the *minimum* length of the expression ($C(x) \approx K(x)$), the closer the compressor-based distance metrics are to the *normalized information distance*. We investigate, empirically, whether the bitrate actually reflects the classification accuracy. We plot bitrate versus classification accuracy for each compressor in Table 2 on three datasets as shown in Figure 3. We use the net bitrate, which is $(N - n_{\text{extra}})/d$, where $N$ is the length of the compressed bitstream, $n_{\text{extra}}$ is the length of the extra bits, and $d$ is the number of pixels. As we can see, a very strong monotonic decreasing correlation between bitrate and accuracy emerges, with Spearman coefficient [65] $r_s = -0.96$, meaning the lower the bitrate is, the higher the classification accuracy is. This

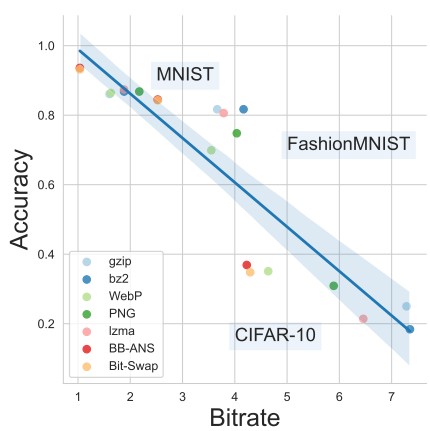

Figure 3: Bitrate versus Classification Accuracy

means the correlation between bitrate and classification accuracy holds empirically regardless of datasets. It will be interesting to investigate in the future whether the correlation remains for lossy compression.

### 6.4 Parallelization and Limitation

The training of generative models can be parallelized using modern GPUs. Compression, however, is not easy to parallelize. The calculation of CDF and PDF are parallelizable for common probability distributions like Gaussian distribution, but the ANS algorithm is not trivial to parallelize. Fortunately, efficient implementations for ANS on GPUs have been developed [66] to exploit GPU's parallelization power. During distance computation stages, as only pair-wise distances are needed, we can use multiple threads to accelerate the computation. For classification, we show that NPC-LV performs well in the low labeled data regime, where the complexity of computation may not be a concern, yet the complexity of $O(n^2)$ may still hinder applications involving pair-wise distance computations like clustering on large datasets, unless we exploit parallelization for compression.

## 7 Conclusion

In this paper, we propose a learning framework, Non-Parametric learning by Compression with Latent Variables (NPC-LV), to address a common learning scenario, Non-Supported Few-Shot-Learning (NS-FSL). This framework is versatile in that every module is replaceable, leading to numerous variations. We use image classification as a case study to demonstrate how to use a trained latent generative model directly for downstream classification without further training. It outperforms supervised learning and non-parametric learning on three datasets and semi-supervised learning on CIFAR-10 in the low data regime. We thus regard it as a baseline in NS-FSL. The equivalence between optimizing latent variable models and achieving the shortest code length not only shows how nELBO can be used for classification, but also indicates that improvements in latent probabilistic models can benefit neural compressors. The relationship between compression rate and classification accuracy suggests that improvements in neural compressors can further benefit classification. Thus, an enhancement of any module in this chain can boost classification accuracy under this framework.

## Acknowledgments

This research was supported in part by the Natural Sciences and Engineering Research Council (NSERC) of Canada, in part by the Global Water Futures program funded by the Canada First Research Excellence Fund (CFREF), and in part by NERC OGP0046506 and Leading Innovative and Entrepreneur teams program of Zhejiang, number 2019R02002, as well as NSFC grant 61832019.

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
