# OpenReview forum: "Few-Shot Non-Parametric Learning with Deep Latent Variable Model"
_NeurIPS.cc/2022/Conference — NeurIPS 2022 Accept_

### Official Review · Reviewer_oRFK · 2022-06-29

**Rating:** 6
**Confidence:** 4
**Soundness:** 3 good
**Presentation:** 3 good
**Contribution:** 3 good

**Summary:**

This paper describes a generative model trained in an unsupervised fashion which utilizes the data distribution to build the compressor. The theory presented in this paper is a minor extension of previous work (see the details below) and the experimental results is not convincing and lack of interpretation and detailed analysis.

**Questions:**

As I pointed out in the weakness of the paper, weak evaluations and lack of interpretations are the main issue of the paper. It will be helpful if the paper can detail the analysis and reveal the insights about why the proposed method can outperform the supervised methods and semi-supervised methods using ablation study. It is also important to make the experimental section more convincing with fair comparison.

Moreover, it is critical to make sure the theory is well connected to the experimental results and support the performance gains.

Last but not least, it is imperative to enhance the uniqueness (novelty) of the paper by explaining the difference of the paper compared to [36] as applying the compression theory to the distance metric in few-shot learning seems to be straight forward and the improvement is incremental.

**Limitations:**

N.A.

**Strengths And Weaknesses:**

The strength of the paper lies in that it proposes a compression-based distance metric and applied it to the few-shot learning.  The experimental results show that the proposed method outperforms some of supervised methods and semi-supervised learning methods on image classification.

The weakness are as follows:

(1)	Compared to the theory proposed in the reference [36], the contribution of this paper seems to be quite incremental as they are all about performing lossless compression with latent variable models. The mathematical framework is largely similar and the intuition behind is the same. The only difference is that in this paper, the compression is applied to the distance metric in few-shot learning setting up which turns out to be a pretty straight-forward extension of [36].
(2)	The experimental results shown in the paper is somewhat not convincing. For instance, the paper claims that the performance of the proposed method is better than supervised learning methods. However, the paper fails to present what components and for what reasons it can outperform supervised learning methods. Because with data samples are fully labeled, supervised learning performances are frequently served as a upper bound of the semi-supervised learning or few-shot learning performances. Without detailed explanations and looking into the insight of the comparison, it is hard to believe the fairness of the comparison.
(namely is the supervised learning methods state-of-the-art, how are the parameter settings of these comparative methods etc)
(3)	The theory presented in this paper are either an extension of the work in [36] or disconnected with the experimental results presented afterward. Namely, I have no idea how to apply these theory to the few shot learning problem and how does these theory support the incremental performance in the experiments. As intuitively, if we compress the distance, we either lose the information and preserve all the information (will not increase the information), so it is not clear where the performance gain comes from. A separate ablation study may help to illustrate the performance gain.

---

> ### Author Response · Authors · 2022-08-02
> **Response to Reviewer oRFK**
>
> We thank the reviewer for the work, but we are afraid that our paper is severely misunderstood. We hope our responses below and Appendix J in the updated paper clarify the scope of our work and its significance.
>
> 1.  We propose NPC-LV, a framework using neural compressors for classification. It consists of a compressor, a distance metric and an aggregation method. As these are all replaceable modules shown in Figure 1, BB-ANS[36] and Bit-Swap[43] here are used _only_ as examples to demonstrate our framework. They can be replaced by other lossless compressors like autoregressive models with arithmetic coding or integer discrete flow. Hence, the major mathematical framework of NPC-LV is _not_ based on [36] or any other specific example of a component, but instead on an information distance derived from Kolmogorov complexity. In fact, we are the first to leverage Kolmogorov complexity based information distance with neural network.
> The intuition behind the information distance is that
> if two objects are similar, there exists a simple program to convert one to another. The "simplicity" of this program characterizes the level of similarity, which is further modeled as Kolmogorov complexity. As Kolmogorov complexity is not computable, the role of a compressor here is to approximate it. We've also added a concrete example of why this framework performs well in Appx J.
>
> 2. Supervised learning methods are not always the upper bound for semi-supervised methods, especially when the amount of training data is limited. This can be validated both empirically and theoretically. Empirically, overfitting prevails in large amounts of previous work when there are limited training samples and complex models (e.g., [72]). Zhang et al.[81] show that even with hundreds of thousands of training samples, deep neural network like CNN still cannot outperform traditional methods like n-gram TFIDF, which is further verified by our result in Appx J. Theoretically, given error tolerance $\epsilon$, training samples $N$ needed can be computed iteratively by: $N\geq \frac{8}{\epsilon^2}\text{ln}(\frac{4((2N)^{d_{vc}}+1)}{\delta})$, where $\delta$ is the confidence parameter and $d_{vc}$ is the VC dimension. Although the bound is loose and $d_{vc}$ only estimates the model capacity instead of the actual parameters, it provides an intuition that the more complex the model is, the more training samples it needs.
>
> 3. We carry out a thorough ablation study in our paper:
> (1) In Table2, we compare non-neural compressors versus neural compressors. We find out that the neural compressors achieve higher accuracy than the non-neural compressors on all three datasets regardless of the distance metric.
> (2) In Table1, we compare the results with and without a compression scheme added to a pretrained latent variable model. We find that the framework with a compression scheme performs better than using latent representation alone.
> (3) In Appx A, we compare four different aggregation methods. We find that to achieve higher accuracy, the aggregation methods are very important and using "average" aggregation method is a simple and effective strategy.
> (4) In Appx A, we also compare latent variables of various numbers. We find that BB-ANS with two latent variables achieves the highest accuracy.
> To further understand where the performance gain comes from, in Section 6.2 we explore the relationship between accuracy and bitrate, and find that a more powerful compressor leads to a better classification accuracy.
>
> 4. To address the concern of the fairness in comparison, we would like to stress that:
> (1) The baselines we choose cover supervised, semi-supervised, non-parametric and NPC methods, each with at least two models.
> (2) We choose the supervised models to cover different model complexity (from 30K parameters to 10M parameters).
> (3) We use the same test and training set for different models.
> (4) For supervised models, we use the validation set to choose the hyperparameters that perform the best (Appx C).
> (5) We run each experiment for 5 times and report mean with 95% confidence interval.
> (6) We report the result of 50-shot, which is beyond our few-shot setting, just to show the situation when our method is inferior to supervised models.
> To further demonstrate that the high performance of the framework is not due to an unfair parameter setting for supervised methods, we show the empirical results of using a simple gzip compressor with completely no training across three _fully_ labeled text datasets.
>
> 	| Model | AGNews | SogouNews | DBpedia |
> 	| :---- | :----: | :-------: | :-----: |
> 	| BiLSTM+Attn| 91.7 | 95.2 | 98.6   |
> 	| HAN   | 89.6  | 95.7     | 98.6   |
> 	| charCNN | 91.4| 95.1     | 98.6   |
> 	| VDCNN | 91.3 | 96.8 | 98.7 |
> 	| NPC (gzip)  | 93.7  | 97.5     | 97.0   |
>
> 	The accuracy(%) of each baseline model can be cross-validated by the previous papers (e.g., [81],[82]). Detailed explanation is updated in the Appx J.

---

> > ### Comment · Reviewer_oRFK · 2022-08-04
> > **Response to Authors**
> >
> > Thanks for the detailed responses to my concerns. The additional experiments and clarifications have addressed the issues. I have updated my rating correspondingly.

---

### Official Review · Reviewer_679F · 2022-07-03

**Rating:** 7
**Confidence:** 3
**Soundness:** 3 good
**Presentation:** 3 good
**Contribution:** 2 fair

**Summary:**

Hi all, I am so very sorry for the incorrect review !!

The authors prropose a strategy for few shot learning by using data compression and nearest neighbor classification.  They focus in particular on a distance based on the kolmogorov complexity. The latter being uncomputable, the authors use different types of compression-based distances.

Given a couple of images, the authors thus compress each individually as well as their aggregation in order to be able to compute the distance.

The authors test their méthodes on multiple datasets : MNIST, FASHIONMNIST and CIFAR10 and show very competitive results.

**Questions:**

What is the effect of the difficulty of the task ? How such a method compete with other representation learning strategies ?

**Limitations:**

Limitations are given and discussed by the authors in section 6.3.

**Strengths And Weaknesses:**

## Strengths
The paper propose a novel method based on theoretical grounds and demonstrate its usability with deep learning. The paper seems theoretically sounds. It is very easy to read and the context is well presented for non expert readers. The experimental section support the contribution with competitive results.

## weaknesses
The datasets are known to be easy discriminative tasks... I think that it woul be interesting to see how the method generalizes with more difficult tasks. A deeper focus on representation learning strategies would help get insight about the benefit of the method.

---

> ### Author Response · Authors · 2022-08-02
> **Response to Reviewer 679F**
>
> We thank the reviewer for the corrected review, clear summary and insightful questions.
>
> 1. The choice of dataset and the shot number are two major factors affecting the difficulty of the task. With regard to the relationship between the difficulty of the task and the performance of our method, we find that our method is more advantageous on more complex dataset with lower shot number in terms of the relative performance. To see that, we calculate the average of the our framework's performance enhancement compared with other methods.
>
> 	|    | 5 shot | 10 shot | 50 shot |
> 	| :-- | :-----:| :------:| :-----:|
> 	|MNIST | 4.8% | 1.9%   | -0.1%    |
> 	|FashionMNIST | 22.4% | 12.3% | 3.9%|
> 	|CIFAR-10 | 55.8% | 38.1% | 6.3% |
>
> 	The lower left part of the table represents higher task difficulty. As the shot number descreases and/or as the difficulty of the task increases, our framework's performance enhancement gets higher.
>
> 	Regarding to the absolute classification accuracy, we think our framework holds its advantage on more compressible datasets as shown in Figure 3. By compressible datasets, we mean those datasets whose probability distribution can be captured by generative models via unlabeled data so that the compression ratio of the datasets can be high (e.g., SVHN, STL10). For datasets like OMNIGLOT, where only small set of unlabeled data is available for each class, it's hard for our framework to perform well. However, with the help of conditional VAE like [2] or different aggregation methods that capture a more profound relationship between images, we may be able to model and approximate the conditional Kolmogorov complexity $K(x_2|x_1)$ directly and thus can tackle datasets like OMNIGLOT.
>
> 2. We compare the representation learnt by VAE with both single latent and hierarchical structure in Table 1. We also add an experiment here to compare features learnt by GAN. Specifically, we train a DCGAN on CIFAR10, using the exact hyper-parameters shown in [1]. We use all the flattened convolutional features of the discriminator from all layers, calculate the euclidean distance and then run kNN. The result is compared with ours below:
>
> 	| Model | 5 shot | 10 shot | 50 shot |
> 	| :---- | :----: | :-------: | :-----: |
> 	| VAE (single) | $17.3\pm 0.9$ | $19.2\pm 0.7$ | $23.4\pm 0.3$ |
> 	| VAE (hier) | $22.2\pm 1.6$ | $24.2\pm 1.9$ | $26.2\pm 2.9$ |
> 	| DCGAN  | $24.7\pm 4.4$  | $28.7\pm 2.1$    | $34.0\pm 2.3$   |
> 	| NPC-LV | $35.3\pm 2.9$ | $36.0\pm 1.8$ | $37.4\pm 1.2$ |
>
> 	We can see on CIFAR10, given the same amount of unlabeled and labeled data, NPC-LV has higher accuracy(%) in non-parametric classification than kNN with features extracted from DCGAN or latent representation learnt by VAE across different shot settings. Given the same dataset, the smaller the labeled dataset is, the higher our relative performance is.
>
> 	To further compare with other representation learning methods, we conduct non-parametric experiments on text datasets. We use a simple compressor gzip under NPC framework and compare with word2vec and sentence BERT. The result is below:
>
> 	| Model | AGNews | SogouNews | DBpedia |
> 	| :---- | :----: | :-------: | :-----: |
> 	| word2vec | 89.2 | 94.3   | 96.1   |
> 	| sentence BERT | 94.0 | 86.0 | 93.7 |
> 	| NPC | 93.7 | 97.5 | 97.0 |
>
> 	Note that NPC with gzip requires _no_ training or pre-training, whereas sentence BERT has been pre-trained on billions of tokens. We can see that with this framework a simple compressor achieves a competitive and even higher result than other representations. A detailed explanation about the experiment is shown in Appendix J.
>
> [1] Radford, Alec, Luke Metz, and Soumith Chintala. "Unsupervised representation learning with deep convolutional generative adversarial networks."
> [2] Hewitt12, Luke B., et al. "The Variational Homoencoder: Learning to learn high capacity generative models from few examples."

---

### Official Review · Reviewer_PPrq · 2022-07-12

**Rating:** 9
**Confidence:** 3
**Soundness:** 4 excellent
**Presentation:** 4 excellent
**Contribution:** 4 excellent

**Summary:**

The paper provides a link between compressed sensing, non-parametric learning, and deep classification networks. By formulating classical compressed sensing results in context of deep learning, the authors propose a learning framework for deep generative modeling using compression, known as Non-Parametric learning by Compression with Latent Variables (NPC-LV). Experiments are conducted on image classification on standard benchmarks (MNIST, CIFAR-10, and Fashion MNIST) and demonstrate compelling results compared to prior work. A strong theoretical analysis support the method.

**Questions:**

In Table 1, for several experiments on MNIST and FashionMNIST, the best performing methods appears to be self-supervised learning and not supervised learning. Why is this the case, and why isn’t supervised learning always achieving top result?


**Limitations:**

Analysis is very thorough and limitations are discussed in section 6.3. Given that one of the limitations is complexity, it would be helpful to report required training/evaluation times and comparison to those of baselines.


**Strengths And Weaknesses:**

The key strength of the paper is the original and novel idea of unification of two streams of research: compressed sensing and classification using deep neural networks. The description of distances and metric learning from compressed sensing and reformulation into DL terminology is extremely helpful and useful. The proposed method demonstrates very compelling results on three image classification benchmarks, outperforming supervised learning and non-parametric learning methods, and semi-supervised learning under low data regime. Theoretical results include derivation of equivalence between optimizing latent variable models and shortest code length, using negative evidence lower bound (nELBO) analysis. Empirically, the authors also show an inverse relationship between accuracy and bitrate.

Overall, the paper is very promising and has few weaknesses. The key limitation seems to be that analysis is focused on VAE and image classification. It would be valuable to understand the relationships of NPC-LV with respect to other popular generative models, e.g., GANs. In computer vision, is is standard practice to learn a good representation, e.g., using VAE or GAN, and re-use the learned space for fine-tuning to many down-stream tasks. In this context, it may be valuable to understand evaluate how good of a representation does NPC-LV learn in comparison to VAE/GAN. However, even with existing analysis and setup, I believe the paper is very promising and would encourage research at the intersection of compressed sensing and DL representation learning, and thus a great contribution to NeurIPS.

---

> ### Author Response · Authors · 2022-08-02
> **Response to Reviewer PPrq**
>
> We thank the reviewer for the thorough feedback and recognition of our work.
>
> 1. For the relationship between NPC-LV and other generative models, we briefly discuss the possibilities of using other generative models at the end of Section 3.2. Theoretically we can use any generative models with density estimation under our framework. For example, BB-ANS / Bit-Swap can be replaced with autoregressive models with arithmetic coding or integer discrete flow. It's not obvious how to use GANs for NPC-LV as GANs do not come with explicit density estimation. We would like to empirically and systematically evaluate NPC-LV with these generative models in the future work.
> 2. To complement the limits of image classification task, we add text classification task with a traditional compressor in Appendix J of the updated paper. We hope the experiments demonstrate the advantage of the general framework and make the evaluation more complete.
>
> 3. As for the representation learning, in Table 1 we compare our result with the latent representation learnt by a single-latent-variable VAE and a hierarchical VAE. They have the exact setting as NPC-LV (i.e. same unlabeled and labeled datasets) and we can see they are inferior to NPC-LV across all the three datasets in every shot, indicating that NPC-LV can better utilize latent variable models in non-parametric classification tasks. To evaluate GAN's performance as representation learning itself, we add another experiment. We train a DCGAN on CIFAR10, using the exact hyper-parameters shown in [2]. We use flattened convolutional features of the discriminator from all layers, calculate the euclidean distance and then run kNN. The result is compared with ours below:
>
> 	| Model | 5 shot | 10 shot | 50 shot |
> 	| :---- | :----: | :-------: | :-----: |
> 	| DCGAN  | $24.7\pm 4.4$  | $28.7\pm 2.1$    | $34.0\pm 2.3$   |
> 	| NPC-LV | $35.3\pm 2.9$ | $36.0\pm 1.8$ | $37.4\pm 1.2$ |
>
> 	On CIFAR10, given the same amount of unlabeled and labeled data, NPC-LV has higher accuracy(%) in non-parametric classification than kNN with features extracted from DCGAN across different shot settings. The advantage is most obvious in 5-shot setting.
>
> 4. We suppose the question is with regard to semi-supervised learning instead of self-supervised learning? Deep neural networks excel when there are abundant labeled data but may not perform well with limited labeled data. That's the exact problem that semi-supervised learning aims to conquer - to design more data efficient algorithms for datasets composed of huge amounts of unlabeled data and scarce labeled ones [1]. Semi-supervised learning methods do need to make a few assumptions to fully take advantage of both labeled and unlabeled data. For example, consistency regularization methods are based on the intuition that perturbation of the data points should not affect the prediction. The hidden assumption is the smooth assumption: if two input points are close by in the input space, they should have the same label. With that intuition, unlabeled data can be utilized to better learn representations encapsulating invariant features, which may be hard for supervised learning to pick up.
> 5. The detailed running time is updated in the last paragraph of Appendix C. To summarize, the pairwise distance computation with two-latent-variable BB-ANS takes about ten hours in 50-shot setting for MNIST and about thirty hours for CIFAR10. For CNN and VGG it takes about half an hour to train a 50-shot labeled data. For VAT ant MT, the training time is much longer than supervised learning and we need to re-train for every shot. For MNIST and FashionMNIST it takes about three hours to run one experiment in a single shot setting and for CIFAR10 it takes about twelve hours.
>
>
> [1] Ouali, Yassine, Céline Hudelot, and Myriam Tami. "An overview of deep semi-supervised learning."
> [2] Radford, Alec, Luke Metz, and Soumith Chintala. "Unsupervised representation learning with deep convolutional generative adversarial networks."

---

> > ### Comment · Reviewer_PPrq · 2022-08-08
> > **Thank you**
> >
> > Thank you for the clarifications and explanations.

---

### Meta-Review · Area_Chair_Xgrh · 2022-08-25

**Recommendation:** Accept
**Confidence:** Certain

**Metareview:**

This paper proposes a learning framework for deep generative modeling called Non-Parametric learning by Compression with Latent Variables (NPC-LV) with a strong theoretical support. The results on image classification benchmarks look compelling and promising. It's impressive that this unsupervised approach achieves strong results even compared with supervised and semi-supervised approaches. The reviewers unanimously think that this work contains novel and interesting ideas to connect the deep generative modeling, non-parametric learning and compressed sensing in an elegant manner. The reviewers also pointed out that more thorough discussion and investigation on generalization ability of the proposed method to other scenarios beyond VAE and image classification would further strengthen the paper. A good discussion on limitations is presented in the paper. After the rebuttal, most concerns that the reviewers raised have been well addressed. The AC recommends acceptance.

**Award:**

Yes

---

### Decision · Program_Chairs · 2022-09-14

Accept